# Induction of Metabolic Changes in Amino Acid, Fatty Acid, Tocopherol, and Phytosterol Profiles by Exogenous Methyl Jasmonate Application in Tomato Fruits

**DOI:** 10.3390/plants11030366

**Published:** 2022-01-28

**Authors:** Silvia Leticia Rivero Meza, Eric de Castro Tobaruela, Grazieli Benedetti Pascoal, Hilton César Rodrigues Magalhães, Isabel Louro Massaretto, Eduardo Purgatto

**Affiliations:** 1Food Research Center, Department of Food Science and Experimental Nutrition, Faculty of Pharmaceutical Sciences, University of São Paulo (USP), Av. Prof. Lineu Prestes 580, bl 14, Butantã, São Paulo 05508-000, SP, Brazil; silvialrmeza@gmail.com (S.L.R.M.); erictobaruela@gmail.com (E.d.C.T.); isabelmassaretto@gmail.com (I.L.M.); 2Faculty of Medicine, Federal University of Uberlândia (UFU), Av. Pará, 1720, bl 2U, Umuarama, Uberlândia 38405-320, MG, Brazil; grazi.nutri13@gmail.com; 3Embrapa Agroindústria Tropical, Rua Dra. Sara Mesquita 2270, Fortaleza 60511-110, CE, Brazil; hilton.magalhaes@embrapa.br

**Keywords:** jasmonate, postharvest management, metabolic changes, tomato fruits, nutritional value, GC-MS

## Abstract

Methyl jasmonate hormone can stimulate the production of several metabolites responsible for improving fruit quality and nutritional attributes related to human health. In this context, efforts to manipulate tomatoes, such as using hormonal treatment to increase metabolite levels essential to plant growth and human nutrition, have received considerable attention. The aim of this study was to show the impact of metabolic profile on fruit quality and nutritional properties under exogenous methyl jasmonate during fruit ripening. The treatments were performed using 100 ppm of methyl jasmonate and 100 ppm of gaseous ethylene over 24 h. Ethylene emission, fruit surface color and metabolomics analysis were measured at 4, 10, and 21 days after harvest, considering the untreated fruits as control group. Methyl jasmonate induced the production of amino acids—mainly glutamine, glutamic acid and γ-aminobutyric acid (at least 14-fold higher)—and fatty acids—mainly oleic, linoleic, and linolenic acids (at least three-fold higher than untreated fruits); while exogenous ethylene predominantly affected sugar metabolism, increasing the levels of fructose, mannose and glucose to at least two-fold that levels in the untreated fruits. Additionally, methyl jasmonate significantly affected secondary metabolites, inducing by at least 80% the accumulation of α-tocopherol and β-sitosterol in fully ripe fruits. Our results suggest that the postharvest application of the hormone methyl jasmonate can contribute to the sensory characteristics and increase the nutritional value of the fruits since important changes related to the tomato metabolome were associated with compounds responsible for the fruit quality and health benefits.

## 1. Introduction

Tomato (*Solanum lycopersicum*) is a valuable fruit crop with economic and nutritional importance. Tomatoes present massive health-promoting compounds such as phenolic compounds, carotenoids, tocopherols and vitamins, which have important biological effects, including antimicrobial, anti-allergenic, anti-inflammatory, antithrombotic, cardio-protective, vasodilatory and antioxidant properties [1]. Several important changes occur during tomato fruit ripening which decisively affect fruit quality and nutritional value. However, recent research has demonstrated that early fruit development plays an essential role in quality parameters such as the accumulation of primary metabolites, including sugars and organic acids [2].

During fruit evolution, morphological stages are evidenced in the tomato fruits, including immature, mature green, breaker, pink and red ripe fruits, which are characterized by the transition from incomplete photosynthetic to complete heterotrophic metabolism [2]. This complex process are coordinated by several phytohormones such as auxins, cytokinins, gibberellins, abscisic acid, salicylic acid, brassinosteroids, ethylene and jasmonates [3]. In addition, recently studies have shown the influence of nitric oxide in controlling plant responses, including the transduction of previously mentioned phytohormones [4,5]. For instance, nitric oxide and ethylene operate antagonistically during the physiological process [6]. Nitric oxide down-regulates fruit ethylene synthesis, suppressing growth, increasing firmness and delaying ripening in tomato fruits [7,8]. In addition, nitric oxide demonstrates that it is a signaling component in MeJA-induced stomatal closure in Arabidopsis guard cells [9].

Methyl jasmonate acts as an important growth regulator and protects tomato plants by inducing biological and biochemical changes [3]. The accumulation of metabolites in plants is associated with the application of exogenous methyl jasmonate, which can stimulate fruit ripening by increasing ethylene production [10,11]. Methyl jasmonate can activate the expression of 1-aminocyclopropane-l-carboxylic acid synthase and 1-aminocyclopropane-l-carboxylic acid oxidase in unripe tomatoes, increasing enzymatic activity and ethylene production [12].

Previous studies have demonstrated that the use of exogenous methyl jasmonate can induce secondary metabolism in plant cells and antioxidant capacity, improving fruit quality [13,14]. However, studies on the effect of methyl jasmonate on the tomato metabolome are mainly related to the production of phenolics, carotenoids, and volatile compounds [10,15]. Minimal knowledge is available concerning a wide variety of metabolites that methyl jasmonate affects during ripening, such as the primary and secondary metabolism. Primary metabolites such as sugars, organic acids, and amino acids are the main classes of chemical compounds responsible for tomato fruit quality, affecting consumers’ sensorial perception [16]. Sugars and organic acids offer sweetness and acid balance, producing a desirable flavor, and amino acids provide taste, such as glutamic acid (umami taste), valine (bitter taste), proline, and serine (sweetness) [17]. Conversely, secondary metabolites play an important role in health promotion, as they are important nutrients for humans. Tocopherols are important antioxidants that scavenge lipid peroxyl radicals and reactive oxygen species in lipophilic environments [18]. Phytosterols such as β-sitosterol are associated with reducing LDL cholesterol and total cholesterol [19], while carotenoids provide the precursor of vitamin A and antioxidant properties [17]. Additionally, some metabolites, such as phenylalanine, tyrosine, tryptophan, linoleic acid, linolenic acid, and lycopene, play important roles as aroma precursors during fruit maturation [20,21].

Thus, the enhancement of fruit quality and the nutritional characteristics of fruits during ripening are commercially essential for horticultural production and are associated with human health. For this reason, the use of methyl jasmonate as an alternative to increasing the production of metabolites can contribute to the sensory and nutritional properties of the crops. Additionally, methyl jasmonate is generally recognized as safe (GRAS) by the Food and Drug Administration, allowing its potential use postharvest. This study investigated the impact of postharvest methyl jasmonate treatment on a wide range of metabolites, including primary and secondary metabolites, of tomato fruits (*Solanum lycopersicum* L. cv. Grape), compared to ethylene treatment and an untreated fruits.

## 2. Results and Discussion

### 2.1. Ethylene Emission and Fruit Surface Color Were Affected by Methyl Jasmonate

The untreated group achieved the breaker stage at 4 DAH, red stage at 10 DAH, and final postharvest stage at 21 DAH. Tomato fruits treated with phytohormones and untreated fruits are shown in Figure 1. Treatments with ethylene and methyl jasmonate accelerated fruit ripening changes to a greater extent than CTRL. The CTRL group showed the peak of ethylene emission at 10 DAH, while ETHY and MeJA showed corresponding peaks at 9 DAH. Interestingly, the level of ethylene emission in MeJA at 9 DAH was at least two-fold higher than that observed in the other groups (Figure 2A). This behavior can be associated with the stimulation of ethylene production in climacteric fruits by the exogenous methyl jasmonate [20].

The phytohormones promoted changes in the tomato surface color slightly faster than in CTRL. ETHY and MeJA first turned red, indicating that treated fruits had completely ripened at 9 DAH, while the CTRL fruits were fully ripe at 10 DAH (Figure 2B). Treatment with ethylene can accelerate color changes in fruits due to its capacity to improve the transcription of mRNAs encoding enzymes, such as phytoene synthase, related to carotenoid production. Methyl jasmonate can also induce carotenoid metabolism in tomatoes [15,22]. Thus, analysis of ethylene emission and surface color revealed that both hormones might play an important role in regulating the carotenoid pathway and lycopene accumulation, accelerating these events in the ripening process.

### 2.2. Impact of Methyl Jasmonate on Primary Metabolites in Tomatoes

Primary metabolites are major components of fruit quality, and related metabolisms are considered crucial for plant growth [23]. Increasing understanding of this process could facilitate developing future strategies for manipulating fruit metabolism. The levels of sugars (fructose, sucrose, and glucose), organic acids (citric and malic acids), amino acids (glutamic acid, γ-aminobutyric acid (GABA), aspartic acid, and aromatic amino acids), and fatty acids (linoleic and linolenic acids) are essential to define the quality of ripe fruits, as they are responsible for the taste of the fruit and used for the synthesis of aroma compounds [20,21]. The sugar metabolism was affected by the ethylene, whereas organic acids, amino acids, and the fatty acid metabolism were impacted mainly by methyl jasmonate (Figure 3). Ethylene evidently modified sugar metabolism during ripening, showing a higher tendency to accumulate fructose, mannose, glucose, sucrose, glucaric acid, allose, gulose, ribose, myo-inositol, and arabinofuranose (Figure 3A). Changes in sugar metabolism in tomato fruits can be related to the interaction of ethylene and nitric oxide. The accumulation of sugars, amino acids and organic acids in fleshy fruits has been associated with action of the nitric oxide, which are important for determining fruit quality as well as flavor and Brix [24].

Nitric oxide plays an important role in the synthesis of ethylene by down-regulating it production and, consequently, delaying fruit ripening by acting as a repressor of ACS and ACO transcript accumulation and enzymatic activity [6,8]. The crosstalk between ethylene and nitric oxide may be associated to the H_2_S and melatonin mediation during fruit ripening [6]. Nitric oxide affects different metabolic pathways, acting early on molecular processes that are responsible for sensory and nutritional quality [24]. For example, several of the genes linked with fruit quality characteristics are modulated in response to nitric oxide in sweet pepper (*Capsicum annuum*) [5]. However, studies related to reactive nitrogen species metabolism and its effects during the ripening of tomato fruit are still scarce [5,24].

The levels of fructose, mannose, and glucose in fruits treated with ethylene were at least two-fold higher than those in the CTRL group. The total sugar level reflected the changes in the most abundant sugars identified as fructose, sucrose, and glucose (Appendix A).

However, methyl jasmonate showed a tendency to increase organic acids; the highest levels of citric, succinic, malic, oxaloacetic, and fumaric acids were detected at 10 DAH (Figure 3A). The levels of citric and malic acids were increased eight-fold, while succinic and oxaloacetic acids were increased to four-fold the levels in the CTRL at 10 DAH. Citric, succinic, and malic acids were essential to the total organic acid level (Appendix A). The remarkable increase in ethylene emission by methyl jasmonate at 9 DAH (Figure 2A) can affect the respiration of tomato fruits, which may explain the increase in the level of organic acids, predominantly in intermediates of the tricarboxylic acid (TCA) cycle, detected mainly at 10 DAH.

Moreover, methyl jasmonate hormone showed a high capacity to accumulate amino acids in the fruits (Figure 3B). Glutamine, glutamic acid, and GABA, free amino acids involved in the GABA shunt, presented levels that were 16-, 2-, and 27-fold higher at 4 DAH and 29-, 27-, and 14-fold higher at 10 DAH, respectively, compared with untreated fruits. Glutamine is known as a relevant form of nitrogen transport in tomatoes, while GABA is a four-carbon non-protein amino acid that has gained considerable attention as a health-promoting functional compound [25]. Thus, the function of free amino acids implicated in the GABA shunt in front of methyl jasmonate treatment deserve more attention owing to their significance in ripening and nutritional quality of fruits. Aromatic amino acids involved in the shikimate pathway were also positively impacted by the postharvest treatment with methyl jasmonate, mainly at 10 DAH. The production of tyrosine increased three-fold, while phenylalanine and tryptophan increased to levels 29-fold higher than those in the CTRL at 10 DAH. The total amino acid level was represented mainly by glutamic and aspartic acids (Appendix A).

The fatty acid profile was significantly affected by the methyl jasmonate hormone. Figure 3C clarifies the tendency of methyl jasmonate to increase the fatty acid metabolism compared with ETHY and CTRL. MeJA showed at least two-fold increases in capric, lauric, myristic, palmitic, stearic, eicosanoic, docosanoic, tricosanoic, lignoceric, cerotic, and montanic acids compared to the values found for untreated fruits during ripening. Unsaturated fatty acids, oleic, linoleic, and linolenic acids, were most impacted by the methyl jasmonate, with their levels increasing by three-fold to four-fold. This is interesting, as they are considered valuable nutrients and important to fruit quality [26]. Palmitic, capric, and eicosanoic acids were essential to the total level of saturated fatty acids and linoleic and oleic acids to the total level of unsaturated fatty acids (Appendix A).

### 2.3. Impact of Methyl Jasmonate on Secondary Metabolites in Tomatoes

Carotenoids, tocopherols, and phytosterols play important roles in health promotion, contributing to the antioxidant capacity of tomato fruits [24]. Lycopene, β-carotene, and lutein were increased by both phytohormones. Major lycopene, β-carotene, and lutein contents (1115.3, 10.16, and 5.25 µg g^−1^ FW, respectively) were detected at 21 DAH in ETHY. The interaction between ethylene and nitric oxide lead not only to antagonism, but also to the promotion of post-translational modifications of antioxidant enzymes. This process results in changes in chlorophyll degradation, lipid peroxidation, pericarp browning, anthocyanin and flavonoid biosynthesis and antioxidant capacity [6]. Nonetheless, the contrary performance was noted at 10 DAH whereas higher lycopene, β-carotene, and lutein contents were revealed in MeJA (742.5, 7.25, and 4.43 µg g^−1^ FW, respectively; Figure 4A and Appendix A). The acceleration of the maturation process evidenced by fruit surface color in ETHY and MeJA (Figure 2B) can be directly correlated to the accumulation of carotenoids, mainly lycopene, compared with the CTRL. Lycopene mainly represented the total carotenoid levels. Ethylene and methyl jasmonate increased lycopene by 40% and 50%, respectively, at 10 DAH and 100% and 40%, respectively, at 21 DAH (Figure 4A).

Tocopherols and phytosterol profiles were affected mainly by methyl jasmonate during ripening (Figure 4B, C). The highest levels of α-, β-, and γ-tocopherol, β-sitosterol, stigmasterol, and stigmastadienol were observed at 10 DAH in MeJA (Appendix A). The production of α-tocopherol, which represented the major source of tocopherol found in Grape tomato, was increased by 85% at 10 and 21 DAH (Figure 4B); β-sitosterol and stigmasterol were important sources of phytosterols identified in tomato fruits. Exogenous methyl jasmonate hormone increased the levels of β-sitosterol and stigmasterol by 80% and 70%, respectively, at 10 DAH, and by 100% and 40%, respectively, at 21 DAH (Figure 4C).

The production of tocopherols, synthesized from phytyl diphosphate generated by geranylgeranyl diphosphate (GGPP) and homogentisic acid from the shikimate pathway [18] in red ripe fruits treated with methyl jasmonate, may be associated with the presence of aromatic amino acids involved in the shikimate pathway, such as phenylalanine, tryptophan, and tyrosin mainly at 10 DAH. Carotenoids and tocopherols share a similar precursor, GGPP, generated by the methylerythritol phosphate pathway. This metabolic interaction indicates that alterations in one of these may influence the biosynthesis of the other metabolite [18]. 

In both hormonal treatments, higher levels of tocopherol were detected at 10 DAH and carotenoids at 21 DAH, indicating that the reduction in tocopherols is matched with a raise in carotenoids. This metabolic interaction suggests that modifications in carotenoid biosynthesis may impact affect the tocopherol level. However, a better comprehension of the accumulation of the mentioned compounds depends on deciphering the complexity of the isoprenoid metabolic network.

### 2.4. Global Overview of Metabolic Changes Occurring in MeJA

This work pointed out the potential of methyl jasmonate application to enhance the levels of primary and secondary metabolites in tomato fruits during ripening off-the-vine in order to improve their quality. However, it is important to highlight the changes in the quality and nutritional value of fruit ripening on the vine. Recent study showed the potential of methyl jasmonate pre-harvest application to increase several phenolic compounds in grapes, suggesting that this phytohormone can be used as elicitor to secondary metabolism [27]. In this context, methyl jasmonate has demonstrated the ability to stimulate the production of a wide variety of metabolites in tomatoes during ripening off-the vine. The entire metabolic pathways were constructed to observe the changes in primary and secondary metabolism in treated fruits compared to CTRL (Figure 5). 

Exogenous methyl jasmonate mainly affected amino acids, fatty acids, tocopherols, and phytosterol profiles. Levels of aspartic acid, asparagine, and threonine were 55-, 12-, and 3-fold higher, respectively, whereas free amino acids, glutamine, glutamic acid, and GABA, increased 29, 27, and 14-fold, respectively, compared to CTRL at 10 DAH. The accumulation of GABA may be related to the largest contents of metabolites being involved in the GABA shunt, such as glutamic acid, succinic acid, and glutamine, mainly at 10 DAH. The increase in the production of asparagine and oxaloacetic acid levels by methyl jasmonate, mainly at 10 DAH, may be responsible for the increase in aspartic acid, reflected in its massive accumulation. Additionally, phenylalanine, tryptophan, and tyrosin were significantly affected by methyl jasmonate, mainly at 10 DAH, suggesting that aromatic acids may contribute to the accumulation of α-tocopherol, which reached levels 85% higher than in untreated red fruits (Figure 4B). However, the highest levels of α-tocopherol were observed after 10 DAH; levels of ethylene and methyl jasmonate hormones at 4 DAH were double those found in untreated fruits (Figure 5). The content of γ-tocopherol contributed only 8% of the total tocopherol level, whereas γ-tocopherol was strongly impacted by methyl jasmonate, increasing its level sixfold compared with untreated fruits (Figure 5).

Both β-sitosterol and stigmasterol were essential sources of phytosterols identified in Grape tomato. Exogenous methyl jasmonate hormone increased the levels of β-sitosterol and stigmasterol by 80% and 70%, respectively, at 10 DAH, and by 100% and 40%, respectively, at 21 DAH (Figure 4C). Fruits treated with exogenous methyl jasmonate showed a strong tendency to accumulate metabolites during their ripening. Generally, the major contents of these metabolites were observed at 10 and 21 DAH. However, the same metabolites were drastically affected by methyl jasmonate at 4 DAH compared with CTRL, as observed for lycopene, α- and γ-tocopherol, β-sitosterol, and stigmasterol, which doubled their contents at the onset of ripening (Figure 5).

The accumulation of metabolites induced by methyl jasmonate treatment leads to acceleration of tomato ripening, which may be associated with the promotion of ethylene biosynthesis by jasmonate [28]. Several studies show the importance of jasmonate in the fruit ripening [29,30,31]. For instance, exogenous application of jasmonate results in increased ethylene production in tomato fruit [32,33]. However, the mechanism involving the increased expression of ethylene signaling genes promoted by jasmonate in tomato is still scarce. In apple fruits, it was observed that ethylene production induced by jasmonate is dependent on the expression of MdACS1 (ACC synthase gene involved in ethylene biosynthesis) and the expression of MdMYC2 (transcription factor involved in the jasmonate signaling pathway) was improved by methyl jasmonate treatment [28].

## 3. Materials and Methods

### 3.1. Fruit Treatment

Mature green tomatoes (*Solanum lycopersicum* cv. Grape; *n* = 1200) were collected from a standard commercial greenhouse in Ibiúna (23°39′21″ S; 47°13′22″ W), São Paulo, Brazil, and transported on ice to the postharvest facilities. Aqueous sodium hypochlorite solution (0.1%) was used for fruit sterilization for 15 min. Four biological replicates were used, each comprising 100 fruits. Fruits were randomly separated into three groups (*n* = 400 by group): (1) control group (CTRL), without treatment; (2) treated ethylene group (ETHY); and (3) treated methyl jasmonate group (MeJA). The treatments were performed using 100 ppm of gaseous ethylene and 100 ppm of methyl jasmonate (Sigma-Aldrich, Saint Louis, MO, USA), considering the final concentration in the gas phase. The methyl jasmonate was applied to a filter paper on the chamber wall for evaporation. The ETHY and MeJA groups were exposed to treatments for the second time 12 h after the first exposure to the hormones, resulting in 24 h of treatment. Fruits were ripened in a 323 L chamber at 20 ± 2 °C and 80 ± 5% relative humidity in a 16-h day/8-h night cycle. Samples of 10 fruits from each replicate were taken at 4, 10, and 21 days after harvest (DAH), considering the control group as reference. The pericarp tissues of tomatoes were frozen in liquid nitrogen and stored at −80 °C for further analysis.

### 3.2. Ethylene Emission

Five fruits were placed in airtight glass containers (600 mL) at 25 °C for 1 h. Five samples of 1 mL of gas produced in the headspace were collected with gastight syringes and injected into a gas chromatograph with a flame ionization detector (GC-FID; Agilent Technologies, Santa Clara, CA, USA, HP-6890) with HP-Plot Q column (30 m × 0.53 mm × 40 µm). The temperatures of the injector and detector were set at 250 °C and the oven at 30 °C. The helium gas flow was set at 1 mL min^−1^, and the injections were performed in pulsed splitless mode.

### 3.3. Fruit Surface Color

The colorimeter (HunterLab ColorQuest XE, Hunter Associates Laboratories, Sunset Hills Road, Reston, VA, USA) measured in terms of *L**, *a**, and *b** space [34]. The fruit surface color was determined at the equatorial zone of six tomato fruits. The results were expressed as the hue angle.

### 3.4. Metabolomic Analysis by GC-MS

#### 3.4.1. Extraction and Derivatization of Polar Metabolites

Frozen pericarp powder (100 mg) was mixed with 100% distilled methanol at −20 °C (1400 μL) and ribitol (200 μg mL^−1^, internal standard; 60 μL). The mixture was vortexed and centrifuged at 11,000× *g* for 10 min. Chloroform at −20 °C (750 μL) and Milli-Q water (1500 μL) were added to the upper phase, followed by centrifugation at 2200 g for 15 min. The upper hydrophilic phase (150 μL) was dried under nitrogen gas. For derivatization, 20 mg mL^−1^ methoxyamine hydrochloride (Sigma-Aldrich Chemical Co., St. Louis, MO, USA; 40 μL) and pyridine were used. *N*-methyl-*N*-(trimethylsilyl) trifluoroacetamide (MSTFA; 70 μL) was added to the sample and incubated in an orbital shaker at 1000 g and 37 °C for 30 min [35]. A pool of polar metabolite external standards (Sigma-Aldrich, Saint Louis, MO, USA) was applied: d-glucose, d-fructose, maltose, sucrose, d-galactose, myo-inositol, citric acid, l-alanine, l-serine, l-proline, l-aspartate, and l-glutamate [36].

#### 3.4.2. Extraction and Derivatization of Non-Polar Metabolites

Frozen pericarp powder (1000 mg) was mixed with chloroform (1250 μL), methanol (2500 μL), *n*-tridecane (800 μg mL^−1^, internal standard; 20 μL). Then, 1.5% sodium sulfate (1250 μL) and chloroform (1250 μL) were added to the mixture, incubated on ice for 5 min, and centrifuged at 4 °C and 1000 *g* for 15 min. The upper polar phase was dried under nitrogen gas and redissolved in hexane (1000 µL), toluene (200 µL), methanol (1500 µL), and 8% chloridric acid (300 µL) and incubated for 1.5 h at 100 °C. The hexane phase was dried under nitrogen gas, redissolved in hexane (80 μL) and pyridine (20 μL), and derivatized with MSTFA (40 μL) [35,36,37]. A pool of fatty acid methyl ester external standards (Sigma-Aldrich, Saint Louis, MO, USA) was used: methyl laurate, methyl tetradecanoate, methyl palmitate, methyl octadecanoate, methyl arachidate, methyl docosanoate, methyl lignocerate, methyl linoleate, (Z)-9-oleyl methyl ester, methyl linolenate, and methyl palmitoleate [38].

#### 3.4.3. Gas Chromatography-Mass Spectrometry Analysis

Derivatized pericarp samples were evaluated by GC-MS (Agilent GC-MS 5977, Agilent Technologies, Santa Clara, CA, USA) with an HP5ms column (30 m × 0.25 m × 0.25 μm) [36]. Trimethylsilyl derivatives (1 μL) were injected into an injector at 230 °C in splitless mode. The oven temperature ramp was 80 °C (initial temperature), held for 2 min, heated at 15 °C min^−1^ to 330 °C, and held for 6 min. The electron impact ionization mass spectrometer was set at ionization voltage 70 eV, ion source temperature 250 °C, injection port temperature 250 °C, and mass scan range 70–600 *m/z* at 20 scans s^−1^. The flow rate of helium gas was 2 mL min^−1^. Experimental data were processed with Mass Hunter software (Agilent, Santa Clara, CA, USA) and validated using the NIST mass spectral library (NIST 2011, Gaithersburg, MD, USA).

#### 3.4.4. HPLC Analysis of Carotenoids

For the extraction of carotenoids, frozen pericarp powder (200 mg) was mixed with 100 μL of 30% NaCl (w:v) solution and 200 μL of dichloromethane. Hexane:diethyl ether (1:1; 500 μL) was added to the mixture and centrifuged (13,000× *g* at 4 °C for 5 min) [39]. The upper phase was dried using nitrogen gas and redissolved in ethyl acetate. The HPLC (Infinity 1260 HPLC, Agilent Technologies, Santa Clara, CA, USA) was coupled to a diode array detector (DAD) equipped with a YMC Carotenoid HPLC C30 (5 µm × 250 mm × 4.6 mm) column [40]. The carotenoid standards used were lycopene, β-carotene, and lutein (Sigma-Aldrich).

### 3.5. Statistical Analysis

Data were expressed as mean ± standard deviation (SD) of four biological replicates. Statistical analysis was performed by ANOVA and Tukey’s test (*p* < 0.05), using the Minitab 19.0 software package. Multivariate and fold change analyses were performed to evaluate the differences between treated and untreated groups using the Metaboanalyst 4.0 server [41]. Raw data were normalized by internal standard area, processed using log transformation (log 2), mean-centered, and divided by the square root of the deviation of each variable (Pareto scaling).

## 4. Conclusions

All these metabolite changes observed during treatment with phytohormone are extremely relevant to studying the fruit quality of tomatoes. However, most of the detected metabolite changes were associated with the sensory and nutritional value, improved by increasing primary and secondary metabolites, respectively. Exogenous methyl jasmonate impacted the profiles of amino acids, fatty acids, tocopherols, and phytosterols by increasing their levels during ripening off-the-vine. Glutamine, glutamic acid, GABA, tryptophan, phenylalanine, and tyrosine were increased by methyl jasmonate, mainly at 10 DAH. Among the fatty acids, oleic, linoleic, and linolenic acids were significantly affected by the methyl jasmonate, increasing their levels by up to four-fold compared with CTRL. Additionally, methyl jasmonate significantly increased the accumulation of α-tocopherol and β-sitosterol in fully ripe fruits. Thus, our results suggest that methyl jasmonate can be used as a tool to contribute to sensory attributes by increasing amino acids and fatty acids, which play an important role in the precursors of volatile compounds and the taste of fruits. Methyl jasmonate also improves the nutritional value of fruits by increasing the accumulation of tocopherols and phytosterols, which are related to health benefits.

## Figures and Tables

**Figure 1 plants-11-00366-f001:**
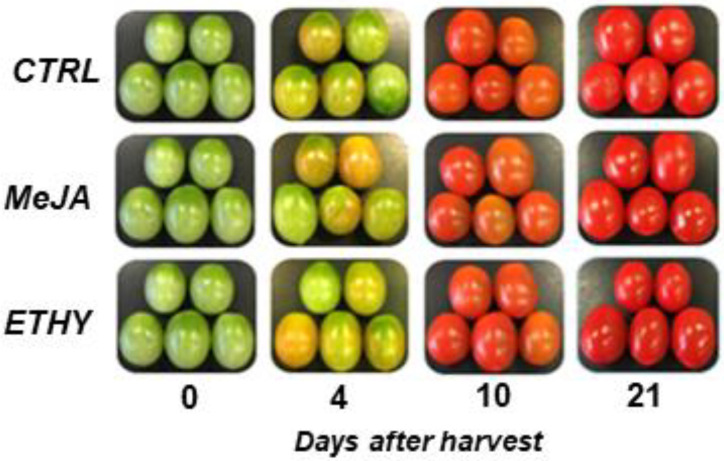
Representative images of tomatoes (*Solanum lycopersicum* L. cv. Grape) exposed to exogenous ethylene (ETHY) and methyl jasmonate (MeJA) compared to the control group (CTRL).

**Figure 2 plants-11-00366-f002:**
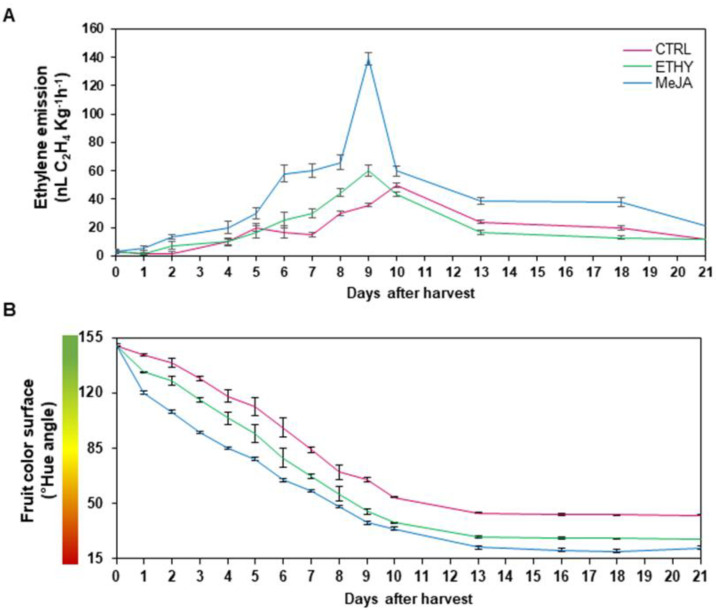
Ethylene emission (**A**) and fruit color (**B**) of tomato (*Solanum lycopersicum* L. cv. Grape) exposed to exogenous ethylene (ETHY) and methyl jasmonate (MeJA) compared to the control group (CTRL). Values are means ± SD of four biological replicates of 10 fruits each.

**Figure 3 plants-11-00366-f003:**
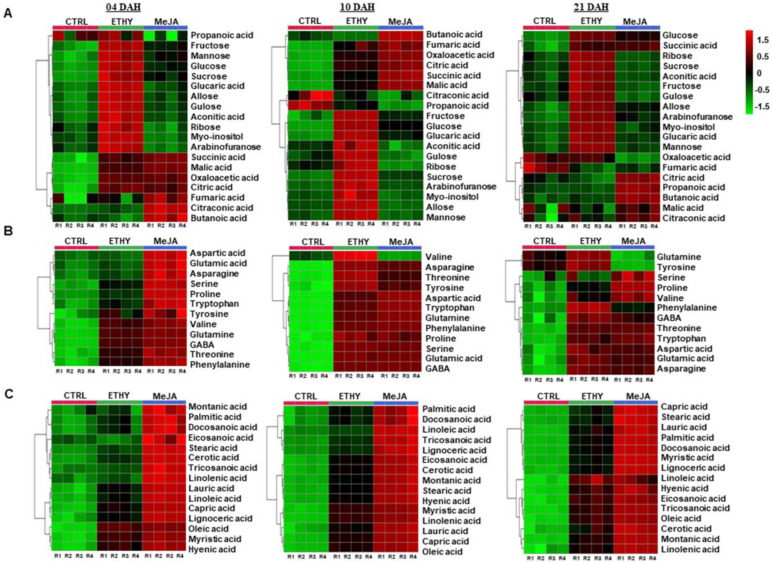
Primary metabolites in tomato (*Solanum lycopersicum* L. cv. Grape) exposed to exogenous ethylene (ETHY) and methyl jasmonate (MeJA) compared to the control group (CTRL). Relative contents of sugars and organic acids (**A**), amino acids (**B**), and fatty acids (**C**). Heatmap analysis representing the major sources of variability. Color scale represents the variation in the relative concentration of compounds, from low (green) to high (red) contents. Relative contents are represented by the normalized area values of each metabolite. Values are means ± SD of four biological replicates of 10 fruits each. Days after harvest (DAH); biological replicates ^®^.

**Figure 4 plants-11-00366-f004:**
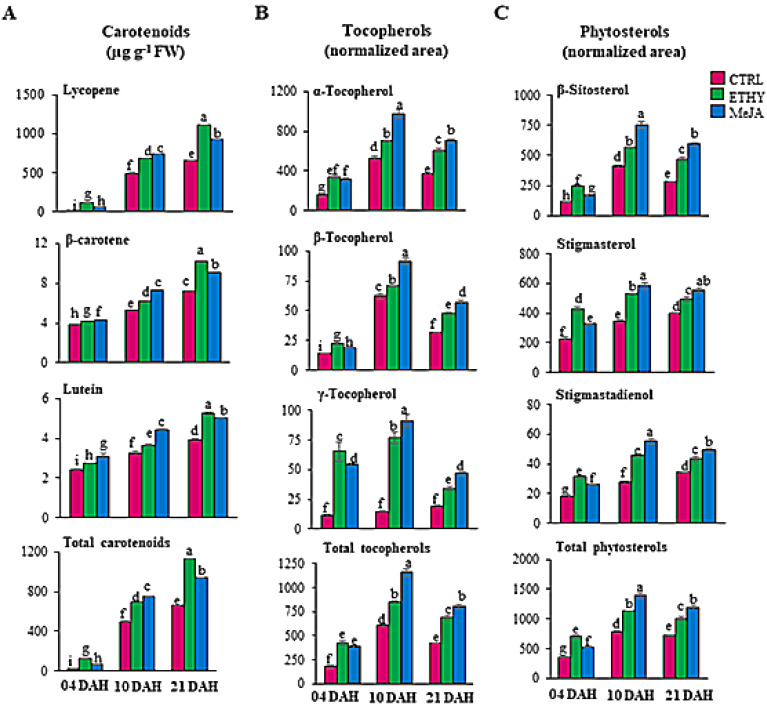
Secondary metabolites in tomato (*Solanum lycopersicum* L. cv. Grape) exposed to exogenous ethylene (ETHY) and methyl jasmonate (MeJA) compared to the control group (CTRL). Contents of carotenoids in µg g^−1^ FW (**A**), relative contents of tocopherols (**B**), and phytosterols (**C**). Heatmap analysis representing the major sources of variability. Color scale represents the variation in the relative concentration of compounds, from low (green) to high (red) contents. Relative contents are represented by the normalized area values of each metabolite. Values are means ± SD of four biological replicates of 10 fruits each. Different letters indicate statistically significant differences (*p* < 0.05). Days after harvest (DAH).

**Figure 5 plants-11-00366-f005:**
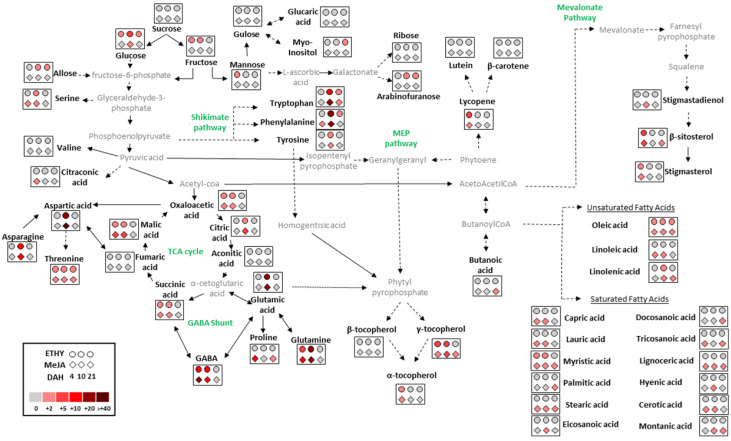
Metabolic changes in tomato (*Solanum lycopersicum* L. cv. Grape) exposed to exogenous ethylene (ETHY) and methyl jasmonate (MeJA) compared to the control group (CTRL). Data were normalized to CTRL. Metabolites presenting up- or down-regulation in each treatment exceeding two-fold compared to CTRL are shown. Color scale is used to display the different amounts of metabolites in terms of the fold change relative to the level in the appropriate control; γ-aminobutyric acid (GABA); tricarboxylic acid (TCA); methylerythritol phosphate (MEP); days after harvest (DAH).

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
