# Peer review of "Induction of Metabolic Changes in Amino Acid, Fatty Acid, Tocopherol, and Phytosterol Profiles by Exogenous Methyl Jasmonate Application in Tomato Fruits"

_plants, 2022, doi:10.3390/plants11030366_

Round 1

Reviewer 1 Report

This is a well written paper. The English is good and the data are presented in a clear way. Thus one can quickly read the paper and understand it.
The experiments presented have been done in a good way, and the data appear to be reliable
This is a useful addition to the literature. My view is the paper is suitable for publication in Plants after minor revision as outlined below
I recommend in the abstract the introduction of concrete results obtained
I recommend in the introduction a brief description of tomato fruits and the importance of the compounds pursued in this study.

Reviewer 2 Report

The manuscript of the research article “Induction of metabolic changes in amino acid, fatty acid, tocopherol, and phytosterol profiles by exogenous methyl jasmonate application in tomato fruits” by Rivero Meza et al. submitted to Plants refers to the topic of enhancement of tomato fruit quality by exogenous methyl jasmonate during ripening off-the-vine. The paper is not very innovative but it is well structured and informative. The experiments are well planned, the results are clearly presented. The methods are modern, adequate and well described. Generally, the statistical methods are relevant and also well described. The biggest disadvantage of this publication is the almost complete lack of discussion of the results. The journal Plants is a plant biology journal the peer-reviewed manuscript in its current form is more relevant to food science than plant biology.

Due to the interesting subject and valuable results, the manuscript can be accepted to publish in Plants but after major revision.

Major comments

  1. The authors should take into consideration the role of nitric oxide in fruit ripening (introduction/discussion). Nitric oxide interacts with the ethylene and jasmonate transduction pathways. I recommend the papers of the Javier Corpas and Jose Palma (Pepe) research group on sweet pepper and tomato and the papers of the team Luciano Freschi research group on tomato.
  2. In the discussion, the authors should refer to the functioning of entire metabolic pathways, add information from experiments based on molecular biology techniques. Please consider this work: The Jasmonate-Activated Transcription Factor MdMYC2 Regulates ETHYLENE RESPONSE FACTOR and Ethylene Biosynthetic Genes to Promote Ethylene Biosynthesis during Apple Fruit Ripening Tong Li, Yaxiu Xu, Lichao Zhang, Yinglin Ji, Dongmei Tan, Hui Yuan, Aide Wang Plant Cell. 2017 29(6): 1316–1334. doi: 10.1105/tpc.17.00349
  3. The authors analyze the issue of the off-the-vine tomato ripening, please discuss the changes in the quality and nutritional value of fruit ripening on the vine.

Minor remarks the authors will find in the attached file.

Round 2

Reviewer 2 Report

Unfortunately, the authors did not address the main comments and did not develop the discussion. The work in this form cannot be published in a biological journal, as it was mentioned in the first report.
The aim of the work the obtained results are very interesting. The manuscript is well written, but in my opinion, it is not acceptable for publication in its current form.

The authors addressed all minor remarks

The authors addressed all minor remarks

Round 3

Reviewer 2 Report

The authors made a major revision of the manuscript. By completing the discussion, the manuscript became a true biological paper.
I accept the manuscript in its current form and recommend it for publication in Plants.